# Effects of Fixture Configurations and Weld Strength Mismatch on *J*-Integral Calculation Procedure for SE(B) Specimens

**DOI:** 10.3390/ma15030962

**Published:** 2022-01-26

**Authors:** Primož Štefane, Stijn Hertelé, Sameera Naib, Wim De Waele, Nenad Gubeljak

**Affiliations:** 1Faculty of Mechanical Engineering, University of Maribor, 2000 Maribor, Slovenia; primoz.stefane2@um.si; 2Soete Laboratory, EMSME Department, Faculty of Engineering and Architecture, Ghent University, 9000 Ghent, Belgium; stijn.hertele@ugent.be (S.H.); sameera.naib@ugent.be (S.N.); Wim.DeWaele@UGent.be (W.D.W.)

**Keywords:** metal weld, strength mismatch, fracture, plastic correction factors, fixture rollers, *J*-*R* resistance curve

## Abstract

This work presents the development of a *J*-integral estimation procedure for deep and shallow cracked bend specimens based upon plastic *η_pl_* factors for a butt weld made in an S690 QL high strength low alloyed steel. Experimental procedures include the characterization of average material properties by tensile testing and evaluation of base and weld metal resistance to stable tearing by fracture testing of square SE(B) specimens containing a weld centerline notch. *J*-integral has been estimated from plastic work using a single specimen approach and the normalization data reduction technique. A comprehensive parametric finite element study has been conducted to calibrate plastic factor *η_pl_* and geometry factor *λ* for various fixture and weld configurations, while a corresponding plastic factor *γ_pl_* was computed on the basis of the former two. The modified *η_pl_* and *γ_pl_* factors were then incorporated in the *J* computation procedure given by the ASTM E1820 standard, for evaluation of the plastic component of *J* and its corresponding correction due to crack growth, respectively. Two kinds of *J*-*R* curves were computed on the basis of modified and standard *η_pl_* and *γ_pl_* factors, where the latter are given by ASTM E1820. A comparison of produced *J*-*R* curves for the base material revealed that variations in specimen fixtures can lead to ≈10% overestimation of computed fracture toughness *J_Ic_*. Furthermore, a comparison of *J*-*R* curves for overmatched single-material idealized welds revealed that the application of standard *η_pl_* and *γ_pl_* factors can lead to the overestimation of computed fracture toughness *J_Ic_* by more than 10%. Similar observations are made for undermatched single material idealized welds, where fracture toughness *J_Ic_* is overestimated by ≈5%.

## 1. Introduction

The fracture resistance (in the form of a *J*-*R* curve) of the weld metal is an essential input for structural integrity assessments of load bearing welded components according to various fitness for service (FFS) assessment methods (e.g., BS7910 [1], R6 [2] and FITNET [3,4]). Experimental determination of the *J*-*R* curve for welded joints is based on the testing of small, laboratory fracture specimens according to standardized procedures, specified by ISO 15,653 [5] and BS 7448-2 [6]. The former is based on the *J*-evaluation method for homogeneous metallic materials, included in ASTM E1820 [7], which has been extended to weldments with yield strength mismatch ratio *M* in the range 0.5 ≤ *M* ≤ 1.25. Here, *M* is defined as:(1)$$M=\frac{\sigma_{yWM}}{\sigma_{yBM}},$$
where *σ_yWM_* and *σ_yBM_* are all-weld metal and base metal yield strength, obtained by tensile testing. In case *J*-integral is evaluated by a single specimen approach, incremental equations are used where elastic and plastic contributions to the strain energy of the cracked specimen are considered according to standard ASTM E1820 as follows:(2)$$J_{\left(i\right)}=J_{el\left(i\right)}+J_{pl\left(i\right)},$$
where *J*_(*i*)_ is the total *J*-integral while *J*_*el*(*i*)_ and *J*_*pl*(*i*)_ are elastic and plastic components of the total *J*-integral respectively. The present study is focused on the determination of *J* by testing of single edge notched bend (SE(B)) specimens. Therefore, *J* computation equations for SE(B) specimens will be discussed throughout this paper. The elastic component *J*_*el*(*i*)_ in Equation (2) for plane strain is given by:(3)$$J_{el\left(i\right)}=\frac{K_{I\left(i\right)}{}^{2}}{E^{\prime }}=\frac{K_{I\left(i\right)}{}^{2}\left(1-\nu^{2}\right)}{E},$$
where, $E^{\prime }=E/\left(1-\nu^{2}\right)$, *E* is the elastic modulus, *ν* is Poisson’s ratio and *K_I_* is the stress intensity factor (SIF) for mode I crack opening, that is:(4)$$K_{I\left(i\right)}=\frac{P_{\left(i\right)}S}{{\left(BB_{N}\right)}^{1/2}W^{3/2}}\cdot f\left(a_{\left(i\right)}/W\right),$$
where *P*_(*i*)_ and *a*_(*i*)_ are the applied load and crack size in the considered time increment, *W*, *B* and *B_N_* denote width, thickness and net thickness (smaller than *B* if the specimen is side-grooved, otherwise *B* = *B_N_*) of the SE(B) specimen and *S* denotes the span length between the support rollers. The function *f*(*a_i_*/*W*) is a nondimensional factor that depends on crack size *a*_(*i*)_ normalized by specimen width *W*, and is given by:(5)$$f\left(a_{\left(i\right)}/W\right)=\frac{3{\left(\frac{a_{i}}{W}\right)}^{1/2}\left[1.99-\frac{a_{i}}{W}\left(1-\frac{a_{i}}{W}\right)\left(2.15-3.93\left(\frac{a_{i}}{W}\right)+2.7{\left(\frac{a_{i}}{W}\right)}^{2}\right)\right]}{2\left(1+2\frac{a_{i}}{W}\right){\left(1-\frac{a_{i}}{W}\right)}^{3/2}}.$$

The plastic component of the *J*-integral, *J*_*pl*(*i*)_, is evaluated by correlation with the area under the load-plastic displacement curve. An incremental equation for computation of *J*_*pl*(*i*)_ was originally proposed by Zhu et al. [8] and is included in ASTM E1820 in the following form
(6)$$J_{pl\left(i\right)}=\left[J_{pl\left(i-1\right)}+\left(\frac{\eta_{pl\left(i-1\right)}}{b_{\left(i-1\right)}}\right)\left(\frac{A_{pl\left(i\right)}-A_{pl\left(i-1\right)}}{B_{N}}\right)\right]\cdot \left[1-\gamma_{pl\left(i-1\right)}\left(\frac{a_{\left(i\right)}-a_{\left(i-1\right)}}{b_{\left(i-1\right)}}\right)\right].$$

Here, *b* is remaining ligament (*b* = *W* − *a*) and *A_pl_* is the area under the load-plastic displacement curve defined by the incremental trapezoidal integration rule:(7)$$A_{pl\left(i\right)}=A_{pl\left(i-1\right)}+\left[P_{\left(i\right)}+P_{\left(i-1\right)}\right]\cdot \left[V_{pl\left(i\right)}-V_{pl\left(i-1\right)}\right]/2.$$

Here, *V_pl_* denotes the plastic component of the measured displacement, which is the crack mouth opening displacement (CMOD) in this paper. Equations (8) and (9) determine *η_pl_* and *γ_pl_* factors, required to calculate the *J*-integral from the load-CMOD record using Equation (6). The former relates to the area under the load- plastic CMOD curve while the latter relates to the incremental plastic work for crack growth. Both factors are functions that depend on normalized crack size for SE(B) specimens.
(8)$$\eta_{pl\left(i-1\right)}=3.667-2.199\left(\frac{a_{\left(i-1\right)}}{W}\right)+0.437{\left(\frac{a_{\left(i-1\right)}}{W}\right)}^{2}$$
(9)$$\gamma_{pl\left(i-1\right)}=0.131+2.131\left(\frac{a_{\left(i-1\right)}}{W}\right)-1.465{\left(\frac{a_{\left(i-1\right)}}{W}\right)}^{2}.$$

Notably, in fracture toughness testing, CMOD is normally preferred over load line displacement (LLD) because a dedicated CMOD gage is simpler to use in an experiment than a complex LLD gage [9]. Furthermore, a study of Kirk and Dodds [10] provided results which showed that the LLD based *J*-integral estimation procedure gives accurate results for *a*/*W* > 0.3, but inaccurate results for *a*/*W* < 0.3 due to a high sensitivity of the *η_pl_* factor to the strain hardening exponent of the material for SE(B) specimens with shallow cracks. In contrast, the CMOD based *η_pl_* factor is insensitive to the strain hardening for *a*/*W* > 0.05 for SE(B) specimens. Fixtures as devices for accurate locating and reliable support during bending testing should be sized according to the ASTM E1820 standard. According to the mentioned standard, rollers should be free in order to keep a constant loading arm. Fixed rollers can have an influence on results, as will be discussed in this paper.

In the past, studies to improve the fracture testing method, based on SE(B) specimens with *W*/*B* = 2 configuration, have introduced improved *η_pl_* factors [10,11,12]. The solutions included in ASTM E1820, in the form of expressions (8) and (9) for computation of *η_pl_* and *γ_pl_* respectively for SE(B) specimens, were proposed by Zhu et al. [8] based on finite element results published by Donato and Ruggieri [12]. Both expressions are assumed to be accurate for a range of crack sizes 0.25 ≤ *a*/*W* ≤ 0.7 [12], while the range of validity is reduced to 0.45 ≤ *a*/*W* ≤ 0.7 in ASTM E1820. As discussed, ISO 15653 assumes that expression (8) is valid for *M* values in the range 0.5 ≤ *M* ≤ 1.25. However, *M* values of weld joints used for various applications often exceed the limit of 1.25. It is necessary to adopt appropriate *η_pl_* and *γ_pl_* equations in terms of strength mismatch *M* and normalized crack length *a*/*W*, in order to accurately evaluate *J*-*R* curves for such joints.

Several researchers provided *η_pl_* and *γ_pl_* solutions for the evaluation of *J* in fracture testing of strength mismatched weld joints. Kim et al. [13] performed detailed finite element (FE) analyses to obtain *η_pl_* of various specimens (including SE(B)) with *a*/*W* = 0.5 for weld joints with strength mismatch *M* varying between 0.5 and 2.0. The obtained results demonstrated that values of *η_pl_* increase in case of weld strength undermatching (*M* < 1) and decrease in case of weld strength overmatching (*M* > 1), relative to even matching welds (*M* = 1). Furthermore, it was shown that *η_pl_* depends on the weld width. Starting from a very wide weld, the reduction of the weld width results in an increase of *η_pl_* values for undermatching welds, reaching a maximum value at geometry ratio (*W* − *a*)/*H_W_* = 5, where *H_W_* is weld width. The opposite was observed for overmatching welds, where reduction of the weld width resulted in an decrease of *η_pl_* values, reaching a minimum value at geometry ratio (*W* − *a*)/*H_W_* = 2. For very narrow welds with (*W* − *a*)/*H_W_* < 2, the effect of the weld geometry was negligible and computed values of *η_pl_* were similar to the one for pure base metal. Eripret and Hornet [14] performed a parametric FE study of SE(B) specimens from weld joints with mismatch levels *M* = 0.2 and *M* = 2.0 for a wide range of crack lengths (0.1 ≤ *a*/*W* ≤ 0.7). Results demonstrated that *η_pl_* values increase across the entire range of analysed crack lengths by reducing the mismatch factor *M*. Similar observations were made by Donato et al. [15]. Their study showed that scatter of produced *η_pl_* solutions for analysed levels of *M* is relatively small if the crack is located at the central plane of a narrow weld with *H_W_* = 5 mm and relatively high in a wider weld with *H_W_* = 20 mm (SE(B) specimens with *B* = 25.4 mm and *W* = 2*B* = 50.8 mm). While aforementioned studies [13,14,15] incorporated 2D plane strain conditions in parametric FEM for SE(B) specimens, Mathias et al. [16] performed parametric 3D FE analyses of SE(B) specimens with *W*/*B* = 1 and *W*/*B* = 2 for a wide range of crack lengths (0.1 ≤ *a*/*W* ≤ 0.7). Although the SE(B) samples were from an X80 steel welded joint with *M* = 1.18 (according to published yield stresses for base and weld material), *η_pl_* solutions were developed for homogeneous material with various yield strength levels and hardening properties. Results revealed that the produced *η_pl_* solution for SE(B) specimens with *W*/*B* = 2 is in close agreement with the one obtained by Donato [12], while the *η_pl_* solution for SE(B) specimens with *W*/*B* = 1 was considerably lower (approx. 11% for shallow cracks and 25% for deep cracks).

The above mentioned reported effects of SE(B) sample geometry, weld size and weld strength mismatch on the *J* evaluation procedure are the main motivation for this research. The work focuses on the development of n_pl_ solutions for SE(B) specimens, extracted from S690 QL steel weld joints with various mismatch levels M. Welding procedures, determination of weld and base material mechanical properties, fracture testing and the calibration of *η_pl_* and *γ_pl_* functions by parametric FEM are presented in the following paragraphs.

## 2. Experimental Procedures

### 2.1. Materials and Welding

Two types of welded plates, as shown in Figure 1 and Figure 2, were fabricated, joining 25 mm thick high strength low alloyed (HSLA) steel S690 QL plates with 500 mm length and 200 mm width, by metal active gas (MAG) welding, with the purpose of extracting specimens for fracture testing and tensile testing. Parent plates during welding were not fixed. For the first type, the weld groove had been machined to a double V configuration with a bevel angle of 60° and root gap of 2 mm; a commonly used weld configuration in practice. For the second type, the weld groove had been machined to a wide V configuration with bevel angle of 20° and weld root gap of 20 mm. In this case, a 10 mm thick backing strip had been attached to both base metal plates to be joined beneath the weld groove in order to fabricate the weld. Such weld configuration meets the requirements of standard ISO 15792-1 [17] for tensile testing of weld consumables and extraction of corresponding tensile test specimens. Weld consumables Mn4Ni2CrMo (with commercial designation MIG 90) and G4Si1 (with commercial designation VAC 65) have been applied in order to fabricate overmatched (OM) welds with *M* > 1 and undermatched (UM) welds with *M* < 1 respectively.

### 2.2. Tensile Testing

Tensile testing of base and all-weld metals was performed in conformance with ASTM E8/E8M [18], utilizing round bar tensile specimens with neck diameter D = 6 mm and gauge length G = 5D = 30 mm. Tensile specimens were tested on a multipurpose testing machine INSTRON 1255 by applying displacement-controlled loading with a crosshead displacement rate of 0.2 mm/min at room temperature. Three tensile tests were performed for each material. The obtained average tensile properties are presented in Table 1, where *E* is the elastic modulus, *σ_YS_* and *σ_UTS_* are yield strength and ultimate tensile strength respectively and *M* is the mismatch factor defined by Equation (1). The obtained average engineering stress-strain (S-e) curves are presented in Figure 3.

Furthermore, fictitious stress–strain curves were computed by offsetting the S-e curves for OM and UM welded joints along the slope of the linear elastic part in order to achieve mismatch factors *M* = 1.5 and *M* = 0.5 respectively. Experimental and fictitious S-e curves were then implemented in FEM in order to investigate the influence of mismatching on solutions for the *J* computation factors *η_pl_*, *λ* and *γ_pl_*.

### 2.3. Fracture Toughness Testing

Fracture toughness testing of base material and welds was performed by the single specimen test method according to ASTM E1820 [7]. SE(B) specimens with thickness and width *B* = *W* = 20 mm have been tested as shown in Figure 4. Three sets of SE(B) specimens were extracted from sample plates with the double V weld configuration for fracture toughness testing of base material, OM weld and UM weld, as shown in Figure 1. Side surfaces of extracted SE(B) specimens containing a weld were first ground and then etched with a 4 % nitric acid alcohol solution in order to determine the position of the weld center line and the weld fusion lines. Finally, notches and aligned knife edges were fabricated at the weld center by wire electrical discharge machining. Weld SE(B) specimens were notched in the direction of the plate thickness (i.e., obtaining surface cracked welded SE(B) specimens). Such notch configuration proved to be less demanding for fatigue precracking, as surface cracked welded SE(B) specimens (with dimensions *B* = *W*) normally do not exhibit a non-uniform fatigue crack front [19] (maximum relative deviations from computed average fatigue crack length *a*_0_ are reported in Table 2). Therefore, no adapted precracking procedure for modification of residual stresses is required. Sharp cracks were introduced in SE(B) specimens through fatigue precracking with load ratio *R* = 0.1 and applied maximum SIF to elastic modulus ratio *K*_max_/*E* ≤ 1.1 × 10^−4^ m^1/2^.

The tests were performed on a multipurpose testing machine INSTRON 1255 under crosshead displacement control at displacement rate 1 mm/min and room temperature. A fixture system with fixed support and load rollers with diameter of 25 mm was used. CMOD was measured with a dedicated clip gauge mounted onto the specimen surface. Stable crack extension in SE(B) specimens was quantified by combining post-mortem crack measurements using the nine-point method and the normalization data reduction (NDR) method as specified in ASTM E1820. Although the NDR method is normally used for stable crack extension estimation in homogeneous metallic materials [20], a recent study conducted by Tang et al. [21] showed that it can be applied to welds as well. An average difference of less than 10% between the *J*-*R* resistance curves provided by the NDR method and unloading compliance method was reported in the listed study. The overview of tested specimens is provided in Table 2. Fracture testing results are presented as load-CMOD curves in Figure 5. Material resistance to fracture in form of *J*-*R* curves are presented and discussed in detail in Section 6, which includes the effect of various iterations of *η_pl_* and *γ_pl_* functions on the computed *J*-integral.

## 3. Numerical Procedures

### 3.1. Weld Geometry Simplification Procedure

Current fracture assessment procedures adopt an idealized weld geometry with straight fusion lines to represent more complex weld configurations found in engineering applications. A systematic methodology for simplification of an actual V-groove weld with a centerline crack to an idealized weld has been proposed by Hertelé et al. [22,23]. This methodology has been developed for single edge tension (SE(T)) specimens and is based on the analysis of slip-line patterns. Research conducted by Souza et al. [24] showed that the weld simplification methodology proposed by Hertelé et al. [22] is adequate for V-grooved welds with straight fusion lines for various weld strength mismatch levels. However, the proposed methodology fails to produce accurate results in the presence of high levels of weld strength undermatch as the deformation pattern near the crack tip changes significantly. Considering this, the double-V weld was in the scope of this research simplified to have bi-linear fusion lines rather than a square weld cross section geometry consisting of perfectly straight fusion lines (Figure 6).

The geometry of overmatched and undermatched welds was modelled symmetrically with respect to the central vertical plane of the SE(B) specimen. Simplification of the double-V weld geometry has been done through post-processing of digital macrographs of actual welds using the following procedure. First, four distinctive fusion lines were recognized with respect to the position of the weld root; upper right, upper left, lower right and lower left fusion line. Points were then marked along each of the four fusion lines and coordinates of the marked points were extracted. Next, straight fusion lines were fitted to the extracted coordinates using linear regression. Average slopes of upper and lower fusion lines have been computed in order to create a symmetrical simplified geometry of the weld. The width of the weld root has been measured in order to accurately adjust the minimum width of the idealized weld in FEM. Finally, the side surfaces of actual SE(B) specimens have been etched and the vertical position of the weld root was measured prior to fracture testing. The position of the weld root was later transferred to the idealized weld in FEM. This way, an idealized weld for each parametric FEM series was adjusted to match a corresponding SE(B) specimen that underwent fracture testing. The simplified weld geometry is shown in Figure 6, while corresponding dimensions are presented in Table 3.

### 3.2. Numerical Models of Tested Specimens

Detailed nonlinear finite element analyses have been performed using ABAQUS 2018. Plane strain finite element models have been created for a wide range of SE(B) specimens with width *W* = 20 mm, width to thickness ratio *W*/*B* = 1 and span length *S* = 4*W* = 80 mm. The analysis matrix shown in Table 4 includes seven distinctive FEM series of SE(B) specimens containing base material and idealized overmatched and undermatched welds in combination with three different support and load roller setups.

Figure 7a,b show examples of plane strain FEM models for SE(B) specimens with *a*_0_/*W* = 0.5 consisting of base metal and containing a welded joint (similar geometry for overmatched and undermatched welds). All other models have similar features. A conventional mesh configuration having a focused ring of finite elements surrounding a stationary crack with a blunted tip. According to SIMULIA documentation [25], contour integral (that is *J* in this paper) should be accurately evaluated if radius of blunted crack tip is *ρ*_0_ ≈ 10^−3^*r_p_*. Here, *r_p_* is size of plastic zone ahead of the crack tip that is determined according to Irwin [26] as:(10)$$r_{p}=\frac{1}{2\pi }{\left(\frac{K_{I}}{\sigma_{YS}}\right)}^{2},$$
where *K_I_* is SIF that is obtained by post-processing of recorded P-CMOD curves (Figure 5) using 95% secant method as specified in ASTM E399 [27]. Size of plastic zone has been estimated for each tested material using Equation (10). Results presented in Table 5 suggest that average crack tip radius ≈1.5 μm could be modelled in all FEM in order to minimize the influence of geometry and mesh on computed results. However, blunt crack tip radius *ρ*_0_ = 2.5 μm was implemented in analyzed FEM as published studies reported that such stationary crack configuration produces sufficiently accurate results [12,15].

The finite element mesh consisted of CPE8R eight node general purpose plane strain elements with reduced integration. In total, the mesh of the analyzed SE(B) models consisted of 10,580 to 10,810 finite elements, depending on crack length *a*_0_/*W* and weld joint configuration. Symmetry conditions were not implemented so that the FEM models allow the replication of asymmetrical weld positions in future work.

Support and load rollers have been modelled as analytical rigid wire parts in the 2D plane strain finite element model. Boundary conditions have been prescribed in reference points at the center of each roller. Parametric studies included three different setups of support and load rollers, which are shown in Figure 8a–c. The first setup, which replicated the standard setup according to ASTM E1820, included support and load rollers with diameters of *d_L_* = 10 mm and *d_S_* = 8 mm, respectively. Support rollers are free to move in the horizontal direction and are fixed in the vertical direction. The second setup served as a control to investigate the influence of load roller diameter on computed *J*-integral values. It included the same support rollers as the previous setup but the load roller diameter increased to *d_L_* = 25 mm. The third setup replicated the actual setup of support and load rollers used in fracture testing of SE(B) specimens. All rollers had diameter *d_L_* = *d_S_* = 25 mm and support rollers were fixed in vertical and horizontal directions. In all finite element models, the load was introduced in displacement control with a prescribed displacement of magnitude 2 mm to the load roller. Rotations of rollers were fixed in all three setups. Contacts have been established between rigid rollers and the deformable SE(B) specimen with a prescribed coefficient of friction *μ* = 0.1.

An elastic-plastic material model, which adopts *J*_2_ flow theory with conventional von Mises plasticity, has been used to describe the material behavior under imposed loads. Material models of base, OM weld and UM weld metals have been determined on the basis of the experimental and offset stress–strain curves presented in Section 2. Plastic properties of listed materials have been implemented in FEM in form of true stress-true plastic strain curves which consisted of up to 21 data points. Finally, small strain assumptions have been implemented in order to enhance the *J*-integral convergence.

## 4. Evaluation of *η_pl_* and *γ_pl_* Factors

Calibration of the *η_pl_* factor is based on a parametric plane strain elastic-plastic finite element analysis of SE(B) specimens with a stationary crack of various lengths. A method established by Donato et al. [15] is implemented in this work and it consists of the following steps:From the results of FEA, extract applied load *P*_(*i*)_, *J*-integral *J_(i)_*, CMOD_(*i*)_ and LLD_(*i*)_ for each simulation increment. Here, *J*-integral is determined by a contour integral method.Compute the area under the *P*_(*i*)_-*V*_*pl*(*i*)_ curve using Equation (7), where *V*_*pl*(*i*)_ denotes plastic component of CMOD_(*i*)_.Compute the elastic component of the *J*-integral *J_el_*_(*i*)_ using Equations (3)–(5).Compute the plastic component of the *J*-integral *J_pl_*_(*i*)_ using Equation (2). Here, *J_(i)_* is the *J*-integral extracted from the FEA results.Compute the normalized area $A_{pl\left(i\right)}^{\prime }$ under the *P*_(*i*)_-*V_pl_*_(*i*)_ curve by the following equation:
(11)$$A_{pl\left(i\right)}^{\prime }=\frac{A_{pl\left(i\right)}}{b^{2}B\sigma_{YS}}.$$Compute the normalized plastic component $J_{pl}^{\prime }$ of the *J*-integral by the following equation:
(12)$$J_{pl\left(i\right)}^{\prime }=\frac{J_{pl\left(i\right)}}{b\sigma_{YS}}$$Create a plot of $J_{pl\left(i\right)}^{\prime }$ as a function of $A_{pl\left(i\right)}^{\prime }$. Compute the *η_pl_* factor for the analysed SE(B) specimen as the slope of the $J_{pl\left(i\right)}^{\prime }$($A_{pl\left(i\right)}^{\prime }$) plot by linear regression.

Once *η_pl_* values are computed for SE(B) specimens with distinct normalized crack lengths *a*/*W* (in range 0.1 ≤ *a*/*W* ≤ 0.7 in this work), the function *η_pl_*(*a*/*W*) can be determined by polynomial curve fitting of the results of *η_pl_* as a function of *a*/*W*.

In the next step, *γ_pl_* factor is evaluated, based on the framework established in the study of Zhu et al. [8], where functions *η_pl_* and *γ_pl_* of *a*/*W* are related as follows:(13)$$\gamma_{pl}\left(a/W\right)=\lambda \eta_{pl}-1-\frac{b}{W}\left(\frac{\lambda^{\prime }}{\lambda }+\frac{{\eta^{\prime }}_{pl}}{\eta_{pl}}\right).$$

Here, *λ* denotes a geometry function *λ*(*a*/*W*), while *λ*′ denotes the derivative *dλ*(*a*/*W*)/*d*(*a*/*W*). Similarly, components *η_pl_* and *η*′*_pl_* denote the function *η_pl_*(*a*/*W*) and its derivative defined as *dη_pl_*(*a*/*W*)/*d*(*a*/*W*), respectively. The function *λ*(*a*/*W*) is defined as the following ratio:(14)$$\lambda \left(a/W\right)=\frac{V_{pl}}{\Delta_{pl}},$$
where *V_pl_* and Δ*_pl_* denote the plastic components of the CMOD and the LLD, respectively. The procedure for the evaluation of *γ_pl_* has been established on the basis of Equations (15) and (16) and contains the following steps:For each analyzed SE(B) specimen with distinct normalized crack length *a*/*W*, create a linear plot in which the plastic component of the LLD Δ*_pl(i)_* is plotted as function of plastic component of the CMOD *V_pl(i)_*. Compute the geometrical factor *λ_(i)_* as the slope of the Δ*_pl_*_(*i*)_ (*V_pl_*_(*i*)_) plot by linear regression method.Determine the function *λ*(*a*/*W*) by curve fitting of the FEA results of *λ* as a function of *a*/*W* in polynomial form.Compute derivatives of *λ(*a*/*W*)* and *η_pl_*(*a*/*W*) functions by differentiating them by the normalized crack length *a*/*W*:
(15)$$\lambda^{\prime }\left(a/W\right)=\frac{d\lambda \left(a/W\right)}{d\left(a/W\right)}$$
(16)$${\eta^{\prime }}_{pl}\left(\frac{a}{W}\right)=\frac{d\eta_{pl}\left(\frac{a}{W}\right)}{d\left(\frac{a}{W}\right)}.$$Compute values of *η_pl_*, *η*′*_pl_*, *λ* and *λ*′ functions for each analyzed SE(B) specimen and insert them in Equation (12) in order to compute *γ_pl_*.Determine *γ_pl_*(*a*/*W*) by polynomial curve fitting of the computed results of *γ_pl_* as a function of *a*/*W*.

## 5. Numerical Results of Plastic *η_pl_* and *γ_pl_* Factors

### 5.1. Verification of FEM and η_pl_ and γ_pl_ Factors for Base Material

Values of *η**_pl_* were obtained from the results of numerical simulations using the procedure described in Section 4. Values of *J*-integral for contours at 0.5 mm and 2.0 mm ahead of the crack tip (further referred as 0.5 mm contour and 2.0 mm contour respectively) were selected for the computation of the *η**_pl_* factors. The size of the 0.5 mm contour is relatively small and is suitable for the computation of *J*-integral in narrow double V welds, as the contour must be located in homogeneous material at the vicinity of the crack tip. The 2.0 mm contour has been used as the reference contour, since it provides converged values of the *J*-integral and does not interact with concentrated deformation due to load roller contact at a high load level in case of deep cracks with *a*/*W* > 0.6. Convergence analysis of the *J* contour integral was performed for the plane strain model of the base material SE(B) specimen with *a*/*W* = 0.5 and standard configuration of rollers at different load levels. The load levels are (normalized by the limit load *F_y_*): *F*/*F_y_* = 0.53, *F*/*F_y_* = 1 and *F*/*F_y_* = 1.21; the results are shown in Figure 9a–c respectively. The analyses showed that contours close to the crack tip (*y <* 0.1 mm) present inconsistent values of *J*-integral due to severe crack tip blunting and thus deformation of finite elements closest to the crack tip. Values of *J* integral fully converge at a distance 2.0 mm ahead of the crack tip with exception of high load levels exceeding *F*/*F_y_* = 1.21. However, the relative deviation of *J* values for distant contours with respect to the 2.0 mm contour is less than 2%. Further comparison of *J* values for 0.5 mm and 2.0 mm contours showed the deviation between both to be less than 3.2% for all load levels, as presented in Figure 9d. Here, *J* values for the 2.0 mm contour represented reference values for computation of the relative deviation.

Verification of the developed finite element model has been conducted for SE(B) geometries consisting of base material only and implementing the standard configuration of load and support rollers (FEM series 1a). Obtained values of the *η_pl_* factors for various crack lengths in the range 0.1 ≤ *a*/*W* ≤ 0.7 were compared to values were published by Wu et al. [28], Kirk and Dodds [10], Nevalainen and Dodds [29], Kim and Schwalbe [11], Kim et al. [30], Donato and Ruggieri [12] and Zhu et al. [8]. Zhu et al. compared solutions proposed by the other listed references and provided solutions for *η_pl_*, *λ* and *γ_pl_*, that were eventually included in ASTM E1820, by curve fitting of the compared results. In this study, two sets of *η_pl_* values, computed from the *J* values obtained from FEM along 0.5 mm and 2.0 mm contours, were included in the comparison. As demonstrated in Figure 9, both sets of *η_pl_* values closely match the existing solutions. The former showed slightly increased deviations for cracks with normalized length *a*/*W* < 0.3 and *a*/*W* > 0.5, while the latter showed excellent agreement along the entire normalized crack length range 0.1 < *a*/*W* < 0.7. This is due to the fact that convergence of *J* values improves with increasingly distant contours from the crack tip. The relative deviation from *η_pl_* values included in standard ASTM E1820 is less than 5% in both cases. Additionally, it is important to emphasize that *η_pl_* values produced in scope of this research do not exhibit increased variation for cracks with normalized length *a*/*W* < 0.2 as is the case for the solution obtained by Donato and Ruggieri [12], shown in Figure 10. The reason is that *η_pl_* values in this paper were computed as slope of function *J*′_*pl*(*i*)_ (*A*′_*pl*(*i*)_) given by Equations (11) and (12) (presented in Section 4). In contrary, Donato and Ruggieri [12] computed *η_pl_* values as an average of function *η_pl_*(CMOD) values that exceeded rigid exclusion condition *A*_*pl*(*i*)_ ≥ *A*_(*i*)_, where *A*_(*i*)_ is area under P-CMOD curve. As a result, *η_pl_* values that did not fully converge can be included in the computation of the average *η_pl_* value for the given crack length, as shown in Figure 11b,c. Support of this discussion is shown in Figure 11a, where *η_pl_* values computed from the created FEM by method of Donato and Ruggieri [12] follow the existing solution provided by Donato and Ruggieri. Based on the above stated arguments, it is assumed that the created FEM has passed the verification process and produces results that are in line with solutions from published researches and the standard ASTM E1820.

Following the FEM verification, remaining configurations of rollers 1b (standard setup, including oversized rollers with diameter *d_S_* = *d_L_* = 25 mm) and 1c (fixed oversized rollers with diameter *d_S_* = *d_L_* = 25 mm) were included in the investigation. Values of *η_pl_*, computed from *J*-integral that was extracted from 0.5 mm and 2.0 mm contours, are presented in Figure 12. Here, the influence of the boundary conditions, that is, roller setup, on *η_pl_* can be recognized.

First, if the standard diameter of the load roller *d_L_* = 10 mm, implemented in FEM series 1a, is increased to *d_L_* = 25 mm, control FEM series 1b, then the *η_pl_* decreases at maximum 3.1% with respect to the standard solution (ASTM E1820) and 5.4% with respect to the baseline solution 1a, when *J* from the 2.0 mm contour is considered. However, both stated solutions seem to be in close agreement when the normalized crack length is *a*/*W* ≥ 0.6.

Second, *η_pl_* further decreases at maximum 12.0% with respect to the standard solution (ASTM E1820) and 13.3% with respect to the reference solution 1a in case of fixed load and support rollers with diameter *d_S_* = *d_L_* = 25 mm, implemented in FEM series 1c. Again, comparison of both solutions is based on the *J*-integral from the 2.0 mm contour.

Third, *η_pl_* values computed on basis of *J*, extracted from the 0.5 mm contour, deviate from the values computed on basis of *J* that was obtained from 2.0 mm contour due to *J* not being fully converged. The former values deviate from the latter by 2.9%, 3.6% and 2.6 % at most for FEM series 1a, 1b and 1c, respectively.

Eventually, *η_pl_* functions were developed by polynomial least squares curve fitting of the computational results. Proposed solutions are presented in Table 6 and Table 7 for 0.5 mm and 2.0 mm contours, respectively.

The geometry factor *λ* has been determined on the basis of LLD and CMOD according to Equation (14). Figure 13 presents obtained values of the geometry factor *λ* as a function of *a*/*W*. Figure 13 demonstrates that *λ* values, computed for FEM series 1a (standard roller setup) are in close agreement with solution included in ASTM E1820 for *a*/*W* ≥ 0.25. In case of shallower cracks with *a*/*W* < 0.25, computed results deviate from the standard solution. The reason is that Zhu et al. [8] produced the solution for *λ* by curve fitting of the existing results in the range 0.25 ≤ *a*/*W* ≤ 0.7. Further inspection of computed results shows that values of *λ* increase by 8.1% at maximum with respect to the reference solution for FEM series 1a if the load roller diameter is increased from 8 mm to 25 mm (FEM series 1b). However, increasing the support rollers diameter from 10 mm to 25 mm and constraining their degrees of freedom has little effect on *λ* values in case of FEM series 1c as computed values deviate 10.9% at maximum with respect to the reference solution for FEM series 1a. Finally, *λ* functions were developed by polynomial curve fitting of the computed results, using the least squares method. Proposed solutions are presented in Table 8.

The crack growth correction factor *γ* has been determined from computed solutions for *η_pl_* and *λ* functions and their derivatives according to Equation (14). A significant deviation of the computed *γ* value functions from the standard solution is observed (Figure 14). Solutions obtained from FEM series 1c and 1a deviate from the standard solution by a factor of 6.1 to 7.7 at most respectively, where *J* was evaluated from the 2.0 mm contour. Similarly, solutions based on *J*, evaluated from the 0.5 mm contour, deviated from the standard solution by a factor of 7.2 to 8.4 at most. In both cases, solutions obtained from FEM series 1b deviated from the standard solution within the specified ranges. Revision of post-processing procedures revealed that such deviations are due to the combination of *η_pl_*, *η_pl_*′, *λ* and *λ*′ values in Equation (14). The obtained solutions are presented in Table 9 and Table 10 for *J* evaluated from 0.5 mm and 2.0 mm contours.

### 5.2. η_pl_ and γ_pl_ Factors for OM and UM Welds

A postprocessing procedure similar to the one described in the previous paragraph has been applied to results of finite element analyses of welded SE(B) specimens. The FEM model of the welded SE(B) specimens is described in detail in Section 3.2. It is important to emphasize that in this case values of *J* integral were obtained from the 0.5 mm contour only. This compact contour can be entirely located in weld material when the crack tip is located in a narrow weld root, thus meeting the requirements of material homogeneity for computation of the *J* contour integral. One disadvantage is that values of *J*-integral are not fully converged at a distance 0.5 mm ahead of the crack tip. However, they deviate less than 3.2% in comparison with *J* values obtained from the 2.0 mm contour, as reported in previous paragraph, which is considered acceptable for the following.

Additionally, the effect of weld yield strength mismatch variation on fracture behavior of the analyzed SE(B) specimens has been investigated. The actual produced OM and UM weld materials have mismatch ratio *M* = 1.31 and *M* = 0.78 with respect to the base material S690 QL, respectively. Additional mismatch ratios *M* = 1.5 and *M* = 0.5 were investigated by implementing material models of the OM and UM weld materials that were obtained by offsetting true tress-strain curves, as described in Section 2.2.

Again, values of *η_pl_* were computed by the Eta-method described in Section 2.2 and are represented in Figure 15. All simulations implemented a fixed roller setup, which replicated the boundary conditions of the actual performed fracture toughness tests. The computed solution for the base material is plotted as the reference; several observations can be made on basis of the plotted results.

First, the influence of the weld geometry and its mechanical properties on *η_pl_* is clearly demonstrated. Values of *η_pl_* decrease significantly when the crack length *a*/*W* is similar to the distance to the weld root *L_W_*/*W* = 0.36 in case of the OM weld material. The opposite can be observed for UM weld material, where *L_W_*/*W* = 0.42.

Second, altered material models produce *η_pl_* values with pronounced minimum and maximum values when *a*/*W* is similar to *L_W_*/*W* in case of yield strength mismatch *M* = 0.5 and *M* = 1.5 respectively. Computed *η_pl_* values are 12.2% higher at *a*/*W* = 0.36 if an altered material model with *M* = 0.5 is implemented in FEM instead of the actual UM material. Moreover, computed *η_pl_* values are 6.9% lower at *a*/*W* = 0.415 if an altered material model with *M* = 1.5 is implemented in FEM instead of the actual OM material.

Functions of *η_pl_* were obtained by polynomial least squares curve fitting of the computed results. The degree of fitted polynomial functions has been carefully selected in order to improve the coefficient of regression *R*^2^ while avoiding excessive variations that are characteristic for higher degree polynomials. The proposed solutions are presented in Table 11.

Following the analysis of *η_pl_*, the geometry factor *λ* has been computed by inserting CMOD and LLD from FEM results into Equation (14). Figure 16 presents computed solutions, which are in close agreement for deep cracked SE(B) specimens with *a*/*W* ≥ 0.5. Furthermore, the influence of material properties is demonstrated for shallower cracks. Here, values of *λ* are lower for welded joints in comparison with all-base metal specimens, while OM weld configurations produce higher values of *λ* in comparison with UM weld configurations. Computed values of *λ* for OM and UM weld configurations deviate from the one of the base material by 13.5% and 19.1% at most, respectively. However, welds with altered yield strength mismatch *M* = 1.5 and *M* = 0.5 exhibit lower values of *λ* in comparison to the OM weld with *M* = 1.31 by 11.5% and the UM weld with *M* = 0.779 by 10.9% at most, respectively. Finally, *λ* functions that are presented in Table 12, were developed by the least squares method on the basis of the aforementioned results.

Finally, solutions for the crack growth correction factor *γ* have been obtained by Equation (13) and are presented in Figure 17, where the solution for the base material is plotted as a reference. Here, the *γ* factors for OM and UM weld configurations vary significantly throughout the range 0.1 ≤ *a*/*W* ≤ 0.7. The analysis of *η_pl_* and *λ* factors and their derivatives reveal that, in the case of welds, the shape of the *η_pl_* function has a significant impact on the variation of the *γ* function. The computed *γ* functions for OM and UM weld configurations deviate from the base material solution by a factor of 10 and 5.6 at most, respectively. Furthermore, the *γ* factor for welds with altered yield strength mismatch *M* = 1.5 and *M* = 0.5 deviate in comparison with the OM weld configuration with *M* = 1.31 by a factor of 9.0 and the UM weld configuration with *M* = 0.779 by a factor of 17.5 at most, respectively. Finally, *γ_pl_* functions that are presented in Table 13 were developed by the least squares method on the basis of the aforementioned results.

## 6. Application to Fracture Toughness Testing

Fracture toughness tests using the single specimen method according to standard ASTM E1820 were performed as described in Section 2.3. *J*-integral resistance (i.e., *J*-*R*) curves for base material, OM weld and UM weld have been computed using the equations for factors *η_pl_*, and *γ_pl_*, numerically evaluated in the scope of this research (Section 5). For the purpose of comparison, equations for factors *η_pl_*, and *γ_pl_*, included in standard ASTM E1820 were used to produce reference *J*-*R* solutions for the base material and both weld configurations. Computed *J*-*R* curves and relative errors with respect to standard *J*-*R* solutions are shown in Figure 18, Figure 19, Figure 20 and Figure 21. Values of the *J*-integral at crack growth onset *J_Ic_* were determined at the intersection of the *J*-*R* curves with the 0.2 mm blunting line and are presented in Table 14, Table 15 and Table 16.

It should be noted that *J*-*R* curves for the base material (Figure 18 and Figure 19), computed with *η_pl_* and *γ_pl_* that were calibrated on basis of FEM series 1a results, are in agreement with standard *J*-*R* solutions. Here, deviations are less than 4% for the 0.5 mm contour (Figure 18) and less than 2% for the 2.0 mm contour (Figure 19). Similarly, values of *J_Ic_* deviate up to 2.5% and 1.8% with respect to standard solutions for *J*-integral computed by 0.5 mm and 2.0 mm contours, respectively. This indicates that the FEM framework has been properly constructed. Moreover, various boundary conditions (i.e., load and support rollers setup) have an effect on *J*-*R* solutions. If the diameter of the load roller is increased from 8 mm to 25 mm as in computation case 1b, then the following observations can be made. Values of the *J* integral throughout the total crack extension Δ*a* are comparable to solution 1a if *η_pl_* and *γ_pl_*, valid for the 0.5 mm contour *J*-integral are implemented. In contrast, values of *J* decrease up to 4% with respect to standard *J*-*R* solution in case of *η_pl_* and *γ_pl_*, valid for the 2.0 mm contour. Finally, if support rollers are constrained and have diameter increased from 10 mm to 25 mm as in computation case 1c, values of *J*-integral throughout the total crack extension Δ*a* decrease up to 15% and 14% with respect to the standard *J*-*R* solution if *η_pl_* and *γ_pl_*, valid for 0.5 mm and 2.0 mm contours are implemented respectively. Moreover, *J_Ic_* is decreased by 10.7% and 8.9% with respect to the standard solution for the 0.5 mm and 2.0 mm contours respectively. All in all, *J* is overestimated if the standard computational procedure is utilized to postprocess results of a fracture toughness test where the modified roller setup has been implemented.

Similar observations can be made for fracture toughness test results of OM and UM welds, shown in Figure 20 and Figure 21 respectively. Here, the presented *J*-*R* curves were computed for load and fully constrained support rollers with diameter of 25 mm, while *η_pl_* and *γ_pl_* functions, calibrated on the basis of the results of FEM series 2a and 3a for the 0.5 mm contour, have been implemented in the computation. Comparison of *J*-*R* curves based on numerical results with the standard *J*-*R* solutions reveals that *J* values throughout the total crack extension Δ*a* are decreased up to 20% for the OM weld (Figure 20) and up to 10% for the UM weld (Figure 21). Moreover, the corresponding *J_Ic_* values, listed in Table 15 and Table 16, indicate a reduction by 10.4% and 5.4% for OM and UM welds, respectively. Again, *J* values are overestimated if the standard computational procedure is used for the postprocessing of results of the fracture toughness test with the utilized modified roller setup.

## 7. Conclusions

This work addresses the effect of weld strength mismatch and configuration of fixtures on *J* estimation formulas to determine fracture toughness from laboratory measurements using the load-CMOD data of SE(B) specimens. The investigation included fracture toughness testing of standard base material SE(B) specimens and surface cracked SE(B) specimens containing an overmatching or undermatching weld, where the notch and welded joint were aligned with the central plane. A new set of *η_pl_*, *λ* and *γ_pl_* factors for computing *J* from experimental results has been developed for a wide range of crack sizes (0.1 ≤ *a*/*W* ≤ 0.7), levels of weld yield strength mismatch and fixture configurations. Aforementioned factors were developed on the basis of computational results, provided by parametric 2D plane strain finite element analyses. The major conclusions of this study can be summarized as follows:Varying the configuration of fixtures has an effect on *η_pl_*, *λ* and *γ_pl_* factors as demonstrated by parametric plane strain analyses of the base material SE(B) specimen. Values of *η_pl_* decrease by 6.4% if the load roller diameter is increased from *d_L_* = 8 mm to *d_S_* = 25 mm and by 14.2% if load and support rollers diameter is set to *d_L_* = *d_S_* = 25 mm, while the latter are fully constrained. Values of *λ* increase throughout the given range of crack lengths for both fixture modifications. Therefore, it is assumed that CMOD increases for the same LLD, while less work of the applied load manifests in the crack driving force if the fixture is modified.Weld geometry in conjunction with weld material properties has a direct effect on *η_pl_* values. A distinctive decrease of *η_pl_* values can be observed if the crack tip is near the weld root in case of an overmatching weld with *M* = 1.31, while the opposite can be observed in the case of an undermatching weld with *M* = 0.78. Modifying the mismatch to *M* = 1.50 (overmatching weld) and *M* = 0.50 (undermatching weld) further enhances the deviation of *η_pl_* near the weld root. Moreover, weld material properties affect the *λ* factor, where all values are in close agreement when *a*/*W* > 0.35 with the exception of *λ* for modified undermatching weld material with *M* = 0.50. For *a*/*W* < 0.35 the *λ* factors for overmatching and undermatching (original and modified) weld material decrease in comparison with the *λ* for the base material. As a consequence of distinctive shapes of the *η_pl_* and *λ* functions, the *γ* functions oscillate with respect to the standard solution.The computed resistances to stable tearing of the tested SE(B) specimens, expressed in terms of *J*-*R* curves, are highly impacted by the *η_pl_*, *λ* and *γ_pl_* factors. Values of *J* throughout the total crack extension Δ*a* reduce with respect to the standard solution by 15% for the base material, by 11% for the undermatching weld and by 18% for the overmatching weld when the set of correction factors calibrated for the utilized fixture (*d_L_* = *d_S_* = 25 mm, constrained support rollers) is used. This indicates that fracture toughness can be overestimated if the standard *η_pl_*, *λ* and *γ_pl_* factors are applied to the postprocessing of results obtained by fracture toughness testing, where a modified fixture has been utilized.

## Figures and Tables

**Figure 1 materials-15-00962-f001:**
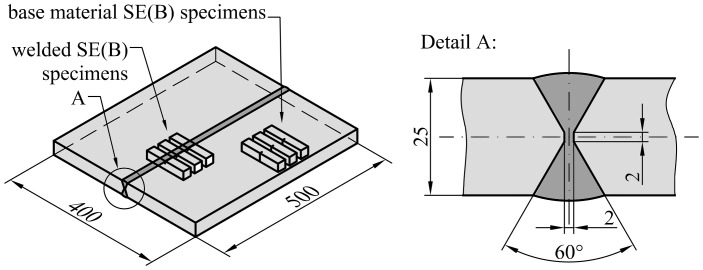
Drawing of the fabricated welded plate for extraction of SE(B) specimens. Weld geometry and layout of the SE(B) specimens is the same for overmatched and undermatched weld.

**Figure 2 materials-15-00962-f002:**
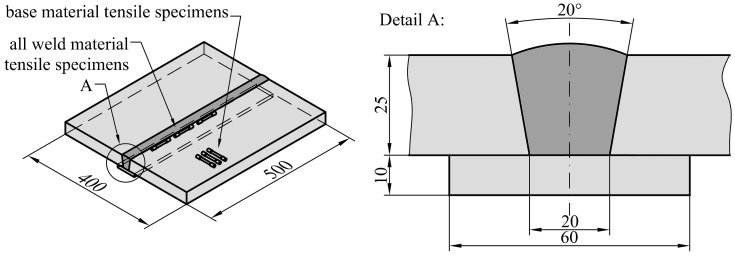
Drawing of the fabricated welded plate for extraction of tensile specimens. Weld geometry and layout of the tensile specimens is the same for overmatched and undermatched weld.

**Figure 3 materials-15-00962-f003:**
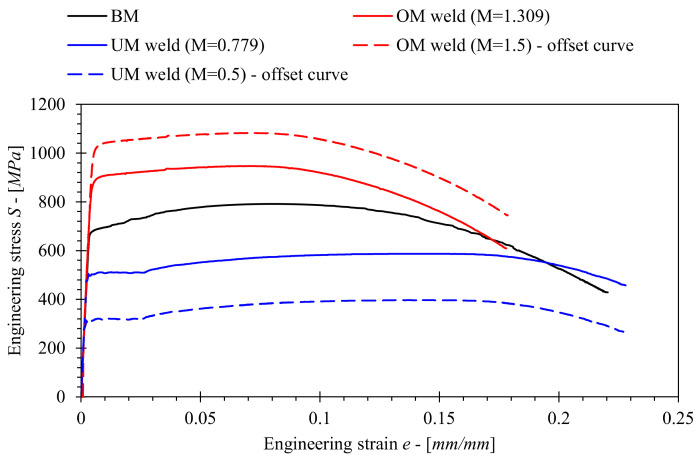
Engineering stress-strain curves of tested materials.

**Figure 4 materials-15-00962-f004:**
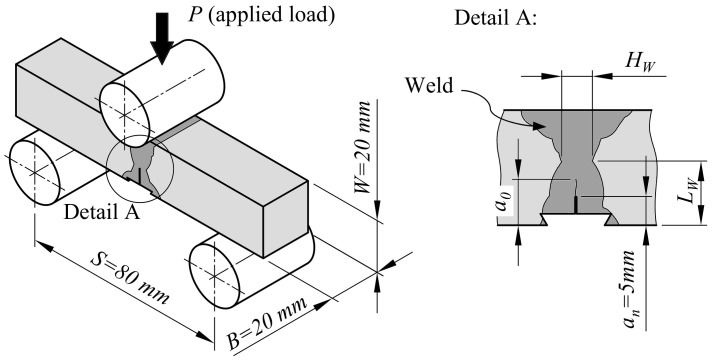
Example of tested weld joint SE(B) specimen with a surface notch at the weld center plane. Missing dimensions *L_W_*, *H_W_* and *a*_0_ are listed in Table 2.

**Figure 5 materials-15-00962-f005:**
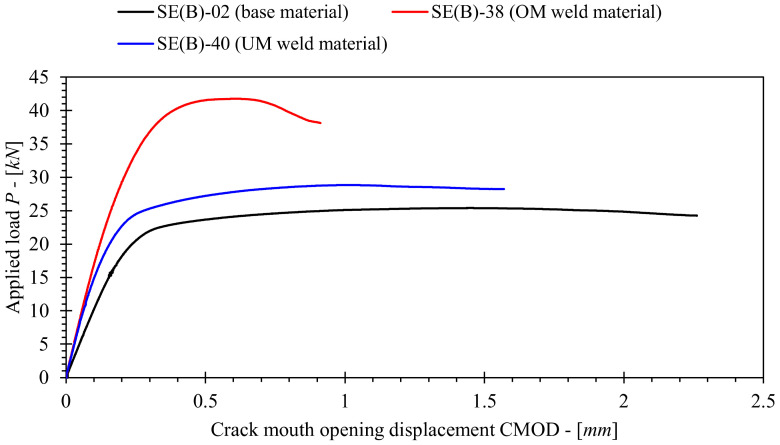
Example of tested weld joint SE(B) specimen with a surface notch at the weld center plane. Missing dimensions *L_W_*, *H_W_* and *a*_0_ are listed in Table 2.

**Figure 6 materials-15-00962-f006:**
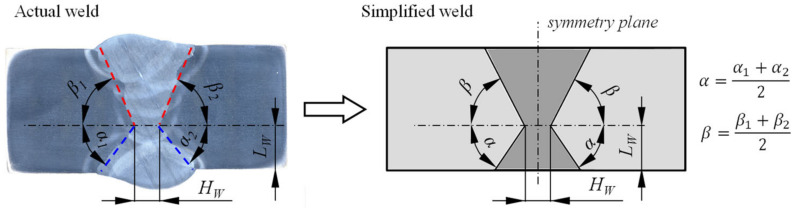
Approach to derive a simplified weld from an irregularly shaped weld.

**Figure 7 materials-15-00962-f007:**
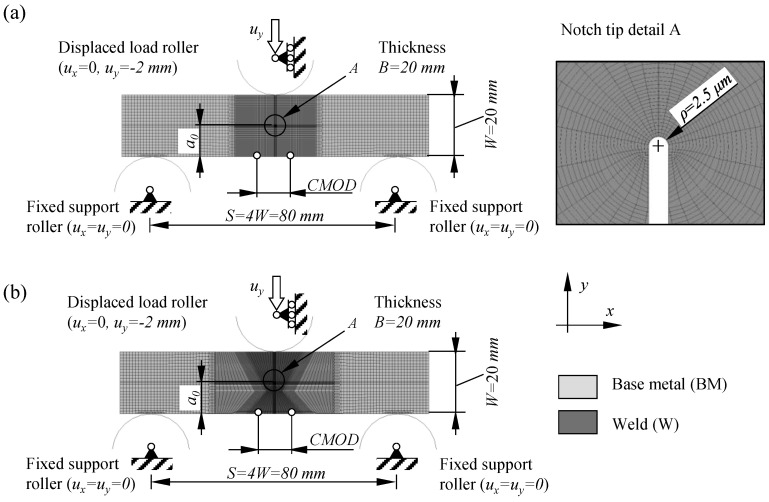
Examples of finite element models of SE(B) specimens: (**a**) containing only base metal and (**b**) containing weld.

**Figure 8 materials-15-00962-f008:**
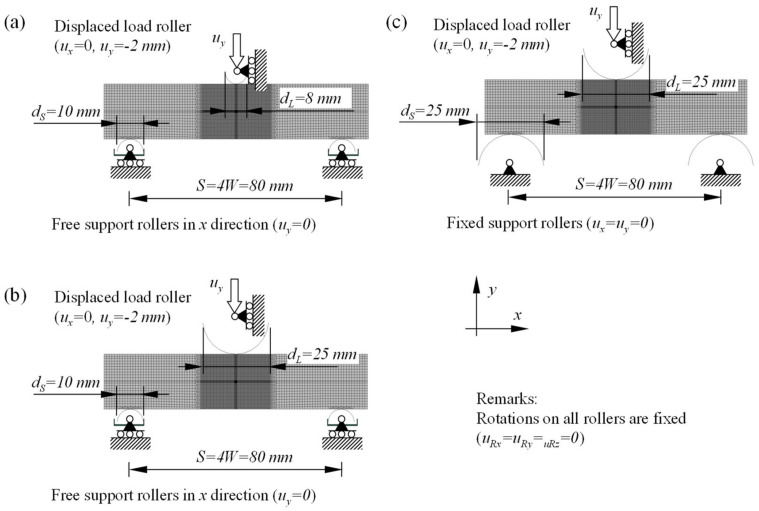
Overview of the investigated load and support rollers setups; (**a**) replica of standardized rollers setup according to ASTM E1820 standard, (**b**) control rollers setup and (**c**) replica of actual rollers setup used in fracture testing.

**Figure 9 materials-15-00962-f009:**
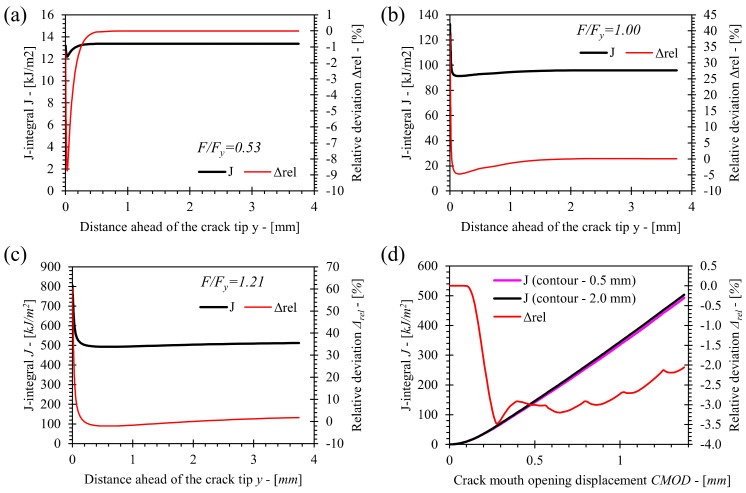
Results of *J*-integral convergence analyses at load levels (**a**) *F*/*F_y_* = 0.53, (**b**) *F*/*F_y_* = 1, (**c**) *F*/*F_y_* = 1.21 and (**d**) comparison of obtained *J*-integral values for 0.5mm and 2.0 mm contours for monotonically loaded FEM of base material SE(B) with *a*/*W* = 0.5.

**Figure 10 materials-15-00962-f010:**
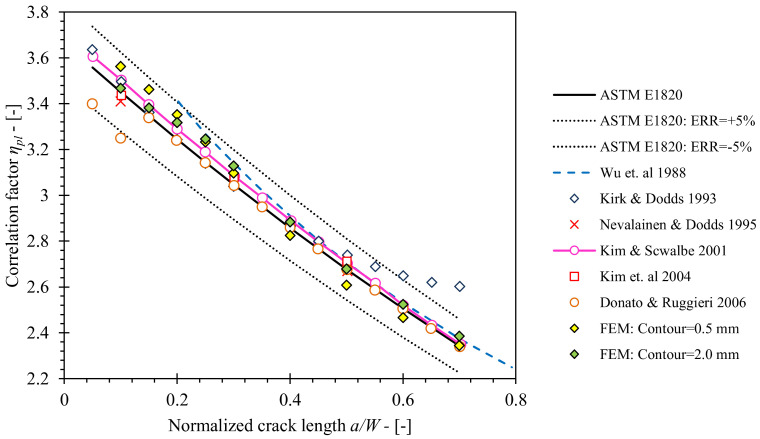
Comparison of *η_pl_* values obtained from the developed FEM framework with the values published in the literature and standard ASTM E1820.

**Figure 11 materials-15-00962-f011:**
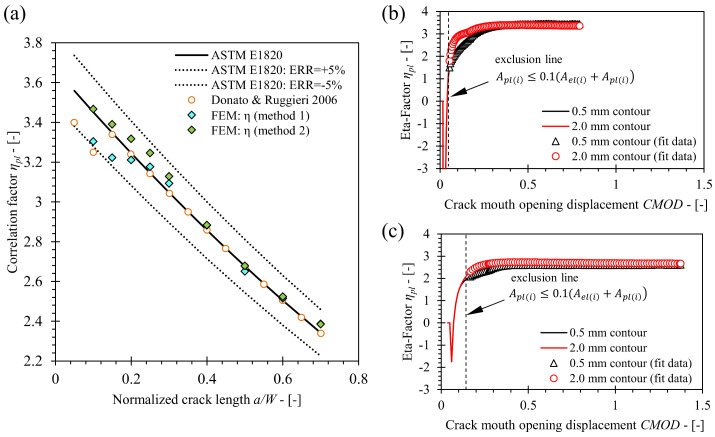
Comparison of (**a**) the current *η_pl_* values computed by method described in Section 4 (denoted as method 1) and method developed by Donato and Ruggieri [12] (denoted as method 2) with the existing solutions. Influence of exclusion criterion on computation of *η_pl_* according to the reference method is demonstrated for SE(B) specimens with crack length (**b**) *a*/*W* = 0.15 and (**c**) *a*/*W* = 0.5.

**Figure 12 materials-15-00962-f012:**
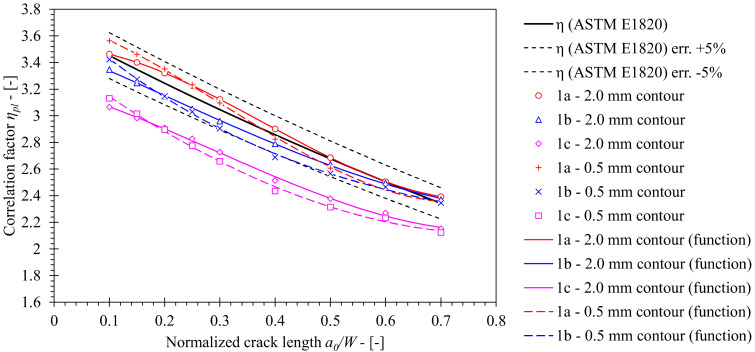
Comparison of *η_pl_* values computed for base material using FEM with standard solution according to ASTM E1820. The former has been computed for numerical models of SE(B) specimens with various fixture and contour configurations.

**Figure 13 materials-15-00962-f013:**
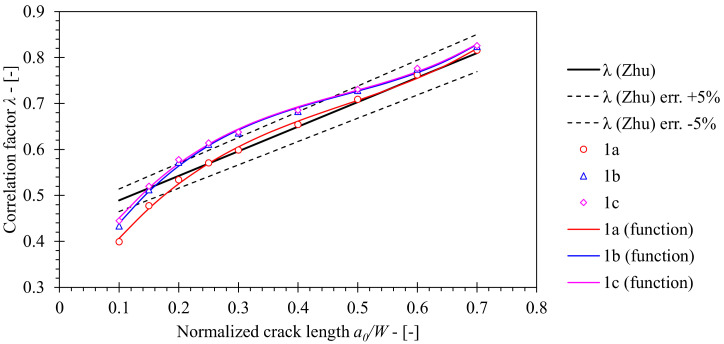
Comparison of *λ* values computed by FEM for base material with standard solution according to ASTM E1820 that was originally proposed by Zhu [8].

**Figure 14 materials-15-00962-f014:**
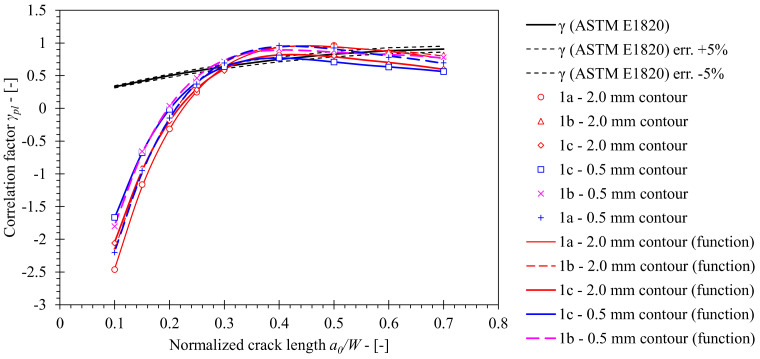
Comparison of the crack growth correction factor *γ* computed for base material using FEM with standard solution according to ASTM E1820. The former has been computed for numerical models of SE(B) specimens with various fixture and contour configurations.

**Figure 15 materials-15-00962-f015:**
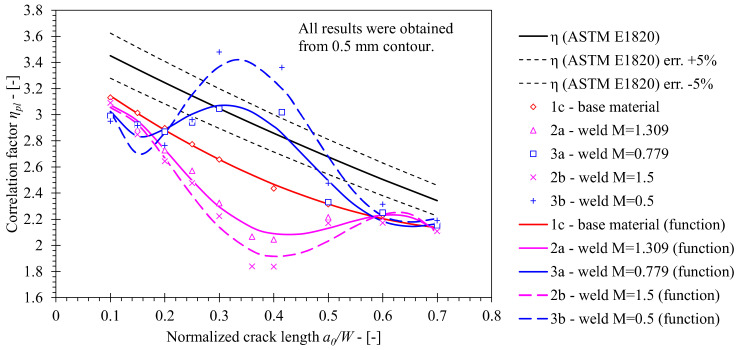
Comparison of *η_pl_* values computed for weld material using FEM with standard solution according to ASTM E1820. The former has been computed for weld materials with various mismatch factors *M*. Solution, obtained for base material, has been plotted as a reference.

**Figure 16 materials-15-00962-f016:**
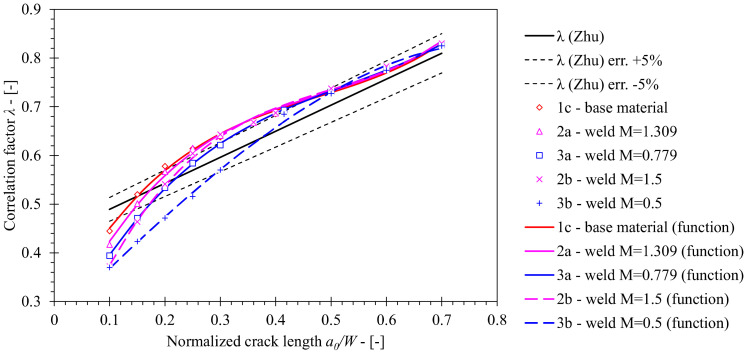
Comparison of *λ* values computed by FEM for weld material with standard solution according to ASTM E1820 that was originally proposed by Zhu [8]. The former has been computed for welds with various mismatch factors *M*. Solution obtained for base material has been plotted as a reference.

**Figure 17 materials-15-00962-f017:**
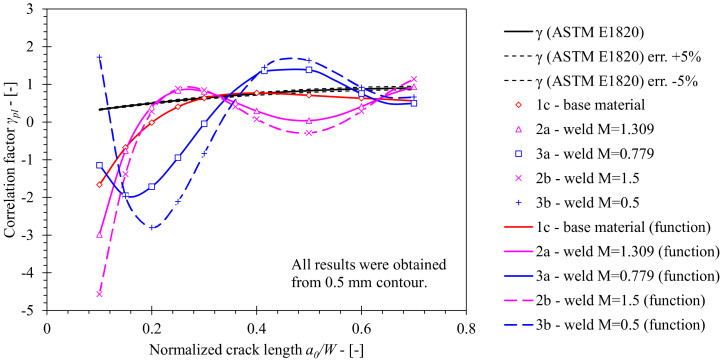
Comparison of the crack growth correction factor *γ* computed for weld material using FEM with standard solution according to ASTM E1820. The former has been computed for welds with various mismatch factors *M*. Solution obtained for the base material has been plotted as a reference.

**Figure 18 materials-15-00962-f018:**
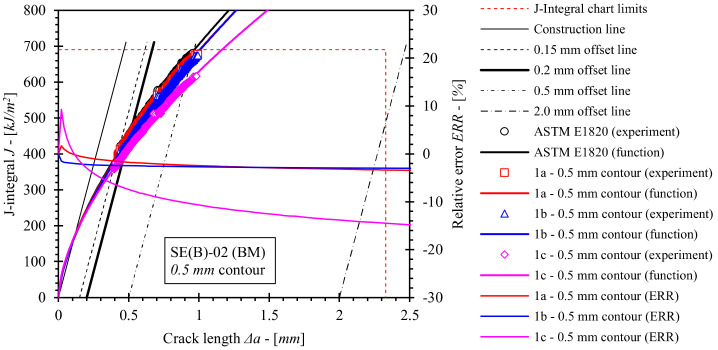
Evaluated *J*-*R* curves for base material, where *η_pl_* and *γ_pl_* functions, evaluated on the basis of numerical results, obtained from the 0.5 mm contour, were implemented. *J*-*R* curves, computed by ASTM E1820 are plotted as a reference.

**Figure 19 materials-15-00962-f019:**
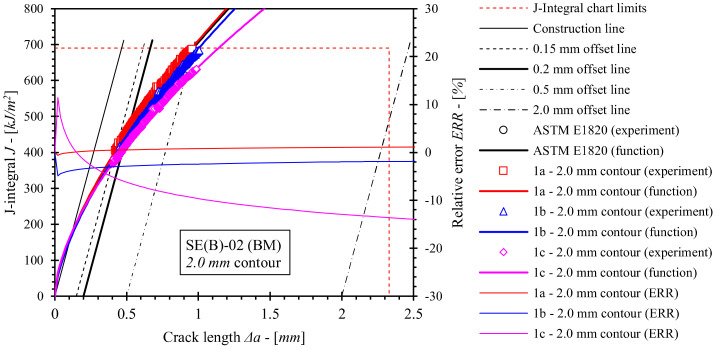
Evaluated *J*-*R* curves for base material, where *η_pl_* and *γ_pl_* functions, evaluated on the basis of numerical results, obtained from the 2.0 mm contour, were implemented. *J*-*R* curves, computed by ASTM E1820 are plotted as a reference.

**Figure 20 materials-15-00962-f020:**
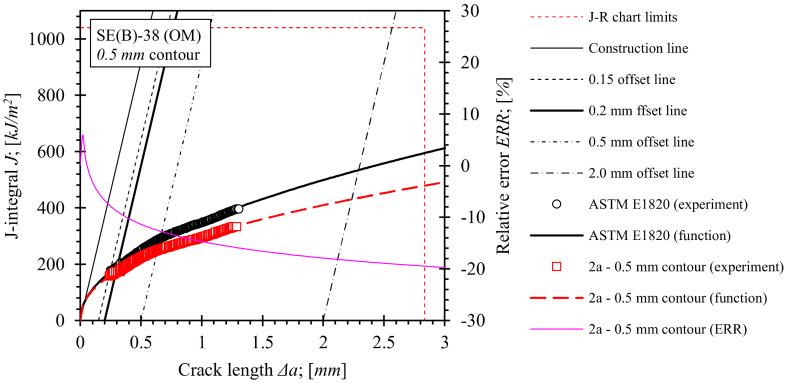
Evaluated *J*-*R* curves for the OM weld. *η_pl_* and *γ_pl_* functions, evaluated on the basis of numerical results, obtained from the 0.5 mm contour, were implemented. *J*-*R* curves that were computed by ASTM E1820 are plotted as a reference.

**Figure 21 materials-15-00962-f021:**
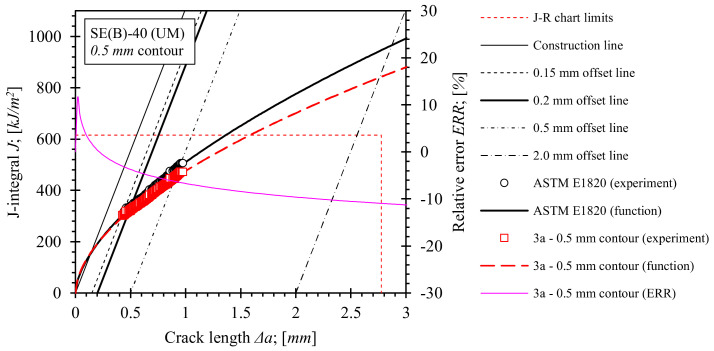
Evaluated *J*-*R* curves for the UM weld. *η_pl_* and *γ_pl_* functions, evaluated on basis of numerical results, obtained from the 0.5 mm contour, were implemented. *J*-*R* curves that were computed by ASTM E1820 are plotted as a reference.

**Table 1 materials-15-00962-t001:** Tensile properties of tested materials.

Material	*E* (GPa)	*σ_YS_* (MPa)	*σ_UTS_* (MPa)	*M* (-)
Base material (S690 QL)	201	683	791	1
OM weld material (Mn4Ni2CrMo)	215	894	950	1.309
UM weld material (G4Si1)	210	532	587	0.779

**Table 2 materials-15-00962-t002:** List of tested SE(B) fracture specimens with corresponding materials and crack lengths.

Specimen	Material	*a*_0_ [mm]	*a*_0_/*W* [-]	*a_f_* [mm]	Δ*a* [mm]	*L_W_* [mm]	*L_W_*/*W* [mm]
SE(B)-02	Base material (S690 QL)	10.645 (7.5%) *	0.533	11.785	1.140	-	-
SE(B)-38	OM weld material (Mn4Ni2CrMo)	8.579 (7.2%) *	0.431	9.888	1.309	7.200	0.360
SE(B)-40	UM weld material (G4Si1)	9.015 (4.2%) *	0.448	9.995	0.980	8.300	0.415

Remarks: *—Maximum relative deviation of 9 measured crack lengths from the average computed crack length (acceptable as specified in ASTM E1820).

**Table 3 materials-15-00962-t003:** Computed dimensions of simplified welds.

Weld	*α* [°]	*β* [°]	*H_W_* [mm]	*L_W_* [mm]
OM weld material (Mn4Ni2CrMo)	48.3	62.7	5.17	7.2
UM weld material (G4Si1)	42.9	51.7	3.47	8.3

**Table 4 materials-15-00962-t004:** The analysis matrix with FEM series distinctive features.

FEM Series	Material	Support and Load Rollers Diameters and Degrees of Freedom	Modelled Normalized *a*_0_/*W* Crack Lengths
1a	base material	*d_S_* = 10 mm (free in horizontal plane) *d_L_* = 8 mm (applied displacement in vertical plane)	0.1, 0.15, 0.2, 0.25, 0.3, 0.4, 0.5, 0.6, 0.7
1b		*d_S_* = 10 mm (free in horizontal plane) *d_L_* = 25 mm (applied displacement in vertical plane)	
1c		*d_S_* = 25 mm (fixed) *d_L_* = 25 mm (applied displacement in vertical plane)	
2a	OM weld *M* = 1.302 (*L*_0_/*W* = 0.36)	*d_S_* = 25 mm (fixed) *d_L_* = 25 mm (applied displacement in vertical plane)	0.1, 0.15, 0.2, 0.25, 0.3, 0.36, 0.4, 0.5, 0.6, 0.7
2b	OM weld *M* = 1.5 (*L*_0_/*W* = 0.36)		
3a	UM weld *M* = 0.779 (*L*_0_/*W* = 0.415)	*d_S_* = 25 mm (fixed) *d_L_* = 25 mm (applied displacement in vertical plane)	0.1, 0.15, 0.2, 0.25, 0.3, 0.415, 0.5, 0.6, 0.7
3c	UM weld *M* = 0.5 (*L*_0_/*W* = 0.415)		

**Table 5 materials-15-00962-t005:** The analysis matrix with FEM series distinctive features.

Material	*K_JIc_* [MPa∙m^1/2^]	*r_p_* [mm]
Base material (S690 QL)	69	1.6 × 10^−3^
OM weld material (Mn4Ni2CrMo)	65	0.9 × 10^−3^
UM weld material (G4Si1)	61	2.1 × 10^−3^

**Table 6 materials-15-00962-t006:** Proposed *η_pl_* functions for SE(B) specimens of base material, valid for *J*, extracted from 0.5 mm contour.

Figure	*η_pl_* Functions for 0.5 mm Contour in Range 0.1 ≤ *a*/*W* ≤ 0.7	*R*^2^ [-]
1a	$\eta_{pl}=4.711{\left(\frac{a}{W}\right)}^{3}-4.339{\left(\frac{a}{W}\right)}^{2}-1.236\left(\frac{a}{W}\right)+3.729$	0.999
1b	$\eta_{pl}=-1.121{\left(\frac{a}{W}\right)}^{3}+3.300{\left(\frac{a}{W}\right)}^{2}-3.789\left(\frac{a}{W}\right)+3.774$	0.999
1c	$\eta_{pl}=0.296{\left(\frac{a}{W}\right)}^{3}+1.556{\left(\frac{a}{W}\right)}^{2}-3.094\left(\frac{a}{W}\right)+3.437$	0.998

**Table 7 materials-15-00962-t007:** Proposed *η_pl_* functions for SE(B) specimens of base material, valid for *J*, extracted from 2.0 mm contour.

FEM Series	*η_pl_* Functions for 2.0 mm Contour in Range 0.1 ≤ *a*/*W* ≤ 0.7	*R*^2^ [-]
1a	$\eta_{pl}=5.025{\left(\frac{a}{W}\right)}^{3}-5.738{\left(\frac{a}{W}\right)}^{2}-0.061\left(\frac{a}{W}\right)+3.552$	0.999
1b	$\eta_{pl}=1.226{\left(\frac{a}{W}\right)}^{3}-0.684{\left(\frac{a}{W}\right)}^{2}-1.751\left(\frac{a}{W}\right)+3.519$	0.998
1c	$\eta_{pl}=2.483{\left(\frac{a}{W}\right)}^{3}-2.181{\left(\frac{a}{W}\right)}^{2}-1.183\left(\frac{a}{W}\right)+3.206$	0.998

**Table 8 materials-15-00962-t008:** Proposed *λ* functions for SE(B) specimens of base material.

FEM Series	*λ* Functions in Range 0.1 ≤ *a*/*W* ≤ 0.7	*R*^2^ [-]
1a	$\lambda =2.278{\left(\frac{a}{W}\right)}^{3}-3.273{\left(\frac{a}{W}\right)}^{2}+2.008\left(\frac{a}{W}\right)+0.236$	0.998
1b	$\lambda =2.806{\left(\frac{a}{W}\right)}^{3}-4.001{\left(\frac{a}{W}\right)}^{2}+2.249\left(\frac{a}{W}\right)+0.252$	0.998
1c	$\lambda =2.603{\left(\frac{a}{W}\right)}^{3}-3.707{\left(\frac{a}{W}\right)}^{2}+2.113\left(\frac{a}{W}\right)+0.274$	0.997

**Table 9 materials-15-00962-t009:** Proposed *γ* functions for SE(B) specimens of base material, valid for *J* extracted from 0.5 mm contour.

FEM Series	*γ* Functions for 0.5 mm Contour in Range 0.1 ≤ *a*/*W* ≤ 0.7	*R*^2^ [-]
1a	$\gamma_{pl}=-65.073{\left(\frac{a}{W}\right)}^{4}+152.019{\left(\frac{a}{W}\right)}^{3}-132.906{\left(\frac{a}{W}\right)}^{2}+50.489\left(\frac{a}{W}\right)-6.048$	1.000
1b	$\gamma_{pl}=-77.924{\left(\frac{a}{W}\right)}^{4}+170.124{\left(\frac{a}{W}\right)}^{3}-137.863{\left(\frac{a}{W}\right)}^{2}+48.739\left(\frac{a}{W}\right)-5.440$	0.999
1c	$\gamma_{pl}=-63.999{\left(\frac{a}{W}\right)}^{4}+144.314{\left(\frac{a}{W}\right)}^{3}-120.653{\left(\frac{a}{W}\right)}^{2}+43.576\left(\frac{a}{W}\right)-4.955$	0.999

**Table 10 materials-15-00962-t010:** Proposed *γ* functions for SE(B) specimens of base material, valid for *J* extracted from 2.0 mm contour.

FEM Series	*γ* Functions for 2.0 mm Contour in Range 0.1 ≤ *a*/*W* ≤ 0.7	*R*^2^ [-]
1a	$\gamma_{pl}=-65.574{\left(\frac{a}{W}\right)}^{4}+152.247{\left(\frac{a}{W}\right)}^{3}-133.564{\left(\frac{a}{W}\right)}^{2}+51.641\left(\frac{a}{W}\right)-6.420$	1.000
1b	$\gamma_{pl}=-73.362{\left(\frac{a}{W}\right)}^{4}+164.154{\left(\frac{a}{W}\right)}^{3}-137.889{\left(\frac{a}{W}\right)}^{2}+51.007\left(\frac{a}{W}\right)-6.032$	1.000
1c	$\gamma_{pl}=-60.373{\left(\frac{a}{W}\right)}^{4}+139.811{\left(\frac{a}{W}\right)}^{3}-121.465{\left(\frac{a}{W}\right)}^{2}+46.027\left(\frac{a}{W}\right)-5.563$	1.000

**Table 11 materials-15-00962-t011:** Proposed *η_pl_* functions for investigated actual and altered weld material SE(B) specimens.

FEM Series	*η_pl_* Functions for 0.5 mm Contour in Range 0.1 ≤ *a*/*W* ≤ 0.7	*R*^2^ [-]
2a	$\eta_{pl}=92.933{\left(\frac{a}{W}\right)}^{5}-269.918{\left(\frac{a}{W}\right)}^{4}+264.133{\left(\frac{a}{W}\right)}^{3}-104.146{\left(\frac{a}{W}\right)}^{2}+13.094\left(\frac{a}{W}\right)+2.569$	0.979
2b	$\eta_{pl}=97.960{\left(\frac{a}{W}\right)}^{5}-316.782{\left(\frac{a}{W}\right)}^{4}+326.940{\left(\frac{a}{W}\right)}^{3}-132.696{\left(\frac{a}{W}\right)}^{2}+17.477\left(\frac{a}{W}\right)+2.339$	0.957
3a	$\eta_{pl}=-380.531{\left(\frac{a}{W}\right)}^{5}+877.582{\left(\frac{a}{W}\right)}^{4}-739.216{\left(\frac{a}{W}\right)}^{3}+274.115{\left(\frac{a}{W}\right)}^{2}-43.820\left(\frac{a}{W}\right)+5.321$	0.934
3b	$\eta_{pl}=-715.044{\left(\frac{a}{W}\right)}^{5}+1647.962{\left(\frac{a}{W}\right)}^{4}-1395.160{\left(\frac{a}{W}\right)}^{3}+522.880{\left(\frac{a}{W}\right)}^{2}-83.327\left(\frac{a}{W}\right)+7.365$	0.880

**Table 12 materials-15-00962-t012:** Proposed *λ_l_* functions for investigated actual and altered weld material SE(B) specimens.

FEM Series	*λ* Functions for 0.5 mm Contour *J*-Integral in Range 0.1 ≤ *a*/*W* ≤ 0.7	*R*^2^ [-]
2a	$\lambda =3.000{\left(\frac{a}{W}\right)}^{3}-4.333{\left(\frac{a}{W}\right)}^{2}+2.442\left(\frac{a}{W}\right)+0.220$	0.997
2b	$\lambda =3.547{\left(\frac{a}{W}\right)}^{3}-5.285{\left(\frac{a}{W}\right)}^{2}+2.972\left(\frac{a}{W}\right)+0.127$	0.998
3a	$\lambda =2.229{\left(\frac{a}{W}\right)}^{3}-3.505{\left(\frac{a}{W}\right)}^{2}+2.254\left(\frac{a}{W}\right)+0.203$	1.000
3b	$\lambda =-0.469{\left(\frac{a}{W}\right)}^{3}-0.141{\left(\frac{a}{W}\right)}^{2}+1.135\left(\frac{a}{W}\right)+0.256$	0.998

**Table 13 materials-15-00962-t013:** Proposed *γ_pl_* functions for investigated actual and altered weld material SE(B) specimens.

FEM Series	*γ**_pl_* Functions for 0.5 mm Contour in Range 0.1 ≤ *a*/*W* ≤ 0.7	*R*^2^ [-]
2a	$\gamma_{pl}=-350.776{\left(\frac{a}{W}\right)}^{4}+691.443{\left(\frac{a}{W}\right)}^{3}-476.044{\left(\frac{a}{W}\right)}^{2}+133.556\left(\frac{a}{W}\right)-12.237$	1.000
2b	$\gamma_{pl}=-501.441{\left(\frac{a}{W}\right)}^{4}+992.155{\left(\frac{a}{W}\right)}^{3}-683.944{\left(\frac{a}{W}\right)}^{2}+191.723\left(\frac{a}{W}\right)-17.844$	1.000
3a	$\gamma_{pl}=-451.632{\left(\frac{a}{W}\right)}^{5}+1422.106{\left(\frac{a}{W}\right)}^{4}-1562.720{\left(\frac{a}{W}\right)}^{3}+734.779{\left(\frac{a}{W}\right)}^{2}-136.680\left(\frac{a}{W}\right)+6.601$	1.000
3b	$\gamma_{pl}=-1503.250{\left(\frac{a}{W}\right)}^{5}+4096.377{\left(\frac{a}{W}\right)}^{4}-4175.130{\left(\frac{a}{W}\right)}^{3}+1940.616{\left(\frac{a}{W}\right)}^{2}-391.926\left(\frac{a}{W}\right)+25.285$	0.999

**Table 14 materials-15-00962-t014:** Computed results of fracture toughness testing of the base material.

Basic Specimen Data	Computation Case, Contour	*J_Ic_* [kJ/m^2^]	*K_JIc_* [MPa∙m^1/2^]
Designation: SE(B)-02 Material: base material (S690 QL) *a*_0_ = 10.645 mm *a*_0_/*W* = 0.533 *a_f_* = 11.785 mm Δ*a* = 1.140 mm	ASTM E1820	448	313
	1a, 0.5 mm	437	309
	1a, 2.0 mm	456	315
	1b, 0.5 mm	421	303
	1b, 2.0 mm	430	306
	1c, 0.5 mm	400	295
	1c, 2.0 mm	408	298

**Table 15 materials-15-00962-t015:** Computed results of fracture toughness testing of the overmatched weld material.

Basic Specimen Data	Computation Case, Contour	*J_Ic_* [kJ/m^2^]	*K_JIc_* [MPa∙m^1/2^]
Designation: SE(B)-38 Material: OM weld material (Mn4Ni2CrMo) *a*_0_ = 8.579 mm *a*_0_/*W* = 0.431 *a_f_* = 9.888 mm Δ*a* = 1.309 mm	ASTM E1820	193	224
	2a, 0.5 mm	173	212

**Table 16 materials-15-00962-t016:** Computed results of fracture toughness testing of the undermatched weld material.

Basic Specimen Data	Computation Case, Contour	*J_Ic_* [kJ/m^2^]	*K_JIc_* [MPa∙m^1/2^]
Designation: SE(B)-40 Material: UM weld material (G4Si1) *a*_0_ = 9.015 mm *a*_0_/*W* =0.448 *a_f_* = 9.995 mm Δ*a* = 0.980 mm	ASTM E1820	336	294
	1a, 0.5 mm	318	285

## Data Availability

Not applicable.

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
