# Peer review of "Effects of Fixture Configurations and Weld Strength Mismatch on J-Integral Calculation Procedure for SE(B) Specimens"

_materials, 2022, doi:10.3390/ma15030962_

Round 1

Reviewer 1 Report

The paper has some merits and the topic is potentially interesting. However, more details descriptions are needed to address. Please consider the following comments.

Comments:

  1. A nomenclature of symbols is necessary.
  2. At the end of Abstract, obtained J-R curves are compared with the results from ASTM E1820, it will be better to add the deviation of the two results.
  3. In Figure 1, the orientation of base material SE(B) specimen is perpendicular to the weldline, the specimen orientation along the weldline is also needed to considered.
  4. In Figure 2, butt welded joint is consisted of weld material, heat affect zone, and base material. So, in the experimental and FE study of SE(B) specimens, the heat affect zone is needed to considered.
  5. In page 5, section 2.3 has the same name with the section 2.1, please correct it.
  6. In Figure 7, the mesh of notch tip detail is used small radius of 2.5 um, the sensitivity analysis of mesh size is necessary to conduct; this is also necessary to consider in the other cases of FE analysis.
  7. In page 10, line 291-297, do different zones of SE(B) specimen adopt different material properties? How to define the fracture criteria?
  8. In page 17, section 5.2, the study of related parameters has also been done for weld materials. However, the weld material is a heterogeneous material, and is the results in section common to use?
  9. The (a) (b) (c)…is needed to add, and capitalize the first letter of words in some figures.

Author Response

Dear Reviewer,

Thank you very much for your very useful suggestions and comments. We would like to stress that many of issues which you have mentioned we are doing now and will be describe in details in future. Our idea is considering weld metal made by two strength different weld metal where crack propagate from over-matching to under-match weld metal.

We have changed a text and description in paper according to your suggestions and marked here as DONE. However, rest of your comments we have cearfuly analysis and provide follows answers:

Your comments:

  1. A nomenclature of symbols is necessary.

Unfortunately, template does not include a nomenclature of symbols. Therefore I am not certain if the this nomenclature must or must not be included.

  1. At the end of Abstract, obtained J-R curves are compared with the results from ASTM E1820, it will be better to add the deviation of the two results. DONE
  2. In Figure 1, the orientation of base material SE(B) specimen is perpendicular to the weldline, the specimen orientation along the weldline is also needed to considered.

Only reported orientations of SE(B) specimens have been analysed so far. SE(B) specimens oriented along the weld line are going to be tested in the future.

  1. In Figure 2, butt welded joint is consisted of weld material, heat affect zone, and base material. So, in the experimental and FE study of SE(B) specimens, the heat affect zone is needed to considered.

This is a part of our research project. Only idealized welds with straight fusion lines and no heat affected zone (HAZ) have been investigated so far in order to establish a baseline solution. Finite element models (FEM) with HAZ are currently being created. In the following weeks, these FEM are going to be processed and results will be analysed.

  1. In page 5, section 2.3 has the same name with the section 2.1, please correct it. DONE
  2. In Figure 7, the mesh of notch tip detail is used small radius of 2.5 um, the sensitivity analysis of mesh size is necessary to conduct; this is also necessary to consider in the other cases of FE analysis. DONE

Strict sensitivity analysis has not been performed. However, we followed certain guidelines when determining blunted notch tip radius;

According to SIMULIA documentation, contour integral (that is J in this paper) should be accurately evaluated if radius of blunted crack tip is ρ0 ≈ 10-3rp, where rp is size of plastic zone ahead of the crack tip. Size of plastic zone ahead of the crack tip rp is determined according to Irwin as rp = (1/2π)*(KI/ σYS), where KI is stress intensity factor obtained post-processing of recorded P-CMOD curves (obtained from fracture tests) using 95% secant method as specified in ASTM E399. Computed rp values along with KI for each tested SE(B) configuration are reported in this paper in Table 5. These results suggest that average crack tip radius ≈ 1.5 μm could be modelled in all FEM in order to minimize the influence of geometry and mesh on computed results. However, blunt crack tip radius ρ0 = 2.5 μm was implemented in analyzed FEM as following studies reported that such stationary crack configuration produces sufficiently accurate results:

  1. H. B. Donato and C. Ruggieri, “Estimation Procedures for J and CTOD Fracture Parameters Using Three-Point Bend Specimens.” pp. 149–157, 25-Sep-2006.

  1. H. B. Donato, R. Magnabosco, and C. Ruggieri, “Effects of weld strength mismatch on J and CTOD estimation procedure for SE(B) specimens,” Int. J. Fract., vol. 159, no. 1, pp. 1–20, 2009.

Furthermore, selected value ρ0 = 2.5 μm is the same orded of magnitude as ≈ 1.5 μm and is therefore assumed that this should not impose a significant error in evaluation of the far field J-integral.

  1. In page 10, line 291-297, do different zones of SE(B) specimen adopt different material properties? How to define the fracture criteria?

Material properties of welds and base material have been obtained by tensile testing of round bar specimens with neck diameter 6 mm. Due to the size, individual tensile specimen includes multiple microstructures of the weld material. Obtained stress-strain curves therefore represent average material properties for multiple microstructures found in the weld. These material properties were then prescribed to the weld region in FEM. This is the most basic simplification of the weld region and is normally used in idealized welds.

Only stationary cracks have been modelled in presented FEM. This is normal procedure when ηpl, λ and γpl functions are being calibrated. Fracture criteria would be important in case of crack propagation. However, establishing a feasible fracture criteria using local approach and prescription of local mechanical properties in the weld is planned for the continuation of this research project.

  1. In page 17, section 5.2, the study of related parameters has also been done for weld materials. However, the weld material is a heterogeneous material, and is the results in section common to use?

Please refer to answer on previous question.

  1. The (a) (b) (c)…is needed to add, and capitalize the first letter of words in some figures. DONE

Kind regards

Nenad

Reviewer 2 Report

The manuscript covers an interesting topic of the development of a J-integral estimation procedure for deep and shallow cracked bend specimens based upon plastic factors for a butt weld made of S690 QL steel. The experimental part of the research is related to the characterisation of material properties by tensile testing and fracture testing of square single edge notch bending specimens. Based on experimental results, J-integral has been estimated., the calibration of plastic factors for various fixture and weld configurations have been conducted by parametric finite element study. Result J-R curves indicate that fracture toughness is overestimated with the standard factors.

This topic is relevant at the international level, and the methodology, analysis, interpretation and organisation are good. The title matches the manuscript content, and relevant literature has been cited.

To improve the manuscript, authors should consider minor changes from specific comments.

Specific comments:

L18-19: Why the lambda factor is not mentioned in the abstract?

L20-21: If possible, the abstract may give a short overview of the conclusions.

L27-28: „…according to various fitness for service (FFS) assessment methods (e.g. BS7910 [1]).“ – is there any other example?

L42-43: „Therefore, J computation equations for the SE(B) specimens will be discussed throughout his paper.“ – this paper?

Figure 7: It is unclear what a) / b) part of the figure is.

Figure 8: It is unclear what a) / b) /c) part of the figure is.

L296-297: „Finally, small strain assumptions have been implemented in order to enhance the J-integral convergence“ – full stop or something missing?

L300, 378, 386, 390: „npl“ – formatting missing

L476: „figure 13“-  it should be figure 14?

L561: „figure 16“-  it should be figure 17?

Figure 18 a): J-R chart limits are not presented

Author Response

Dear Reviewer,

Thank you very much for your very useful suggestions and comments. We would like to stress that many of issues which you have mentioned we are doing now and will be describe in details in future. Our idea is considering weld metal made by two strength different weld metal where crack propagate from over-matching to under-match weld metal.

We have changed a text and description in paper according to your suggestions and marked here as DONE, as you can see.

Your comments:

L18-19: Why the lambda factor is not mentioned in the abstract? DONE

L20-21: If possible, the abstract may give a short overview of the conclusions. DONE

L27-28: „…according to various fitness for service (FFS) assessment methods (e.g. BS7910 [1]).“ – is there any other example? DONE

L42-43: „Therefore, J computation equations for the SE(B) specimens will be discussed throughout his paper.“ – this paper? DONE

Figure 7: It is unclear what a) / b) part of the figure is. DONE

Figure 8: It is unclear what a) / b) /c) part of the figure is. DONE

L296-297: „Finally, small strain assumptions have been implemented in order to enhance the J-integral convergence“ – full stop or something missing? DONE

A: Full stop.

L300, 378, 386, 390: „npl“ – formatting missing DONE

L476: „figure 13“-  it should be figure 14? DONE

L561: „figure 16“-  it should be figure 17? DONE

Figure 18 a): J-R chart limits are not presented DONE

Kind regards

Nenad

Reviewer 3 Report

The paper has a comprehensive presentation of the research results focused on the effects of fixture configurations and weld strength mismatch on J-integral calculation procedure for SE(B) specimens.

The paper has numerous experimental and numerical procedures, FEM calculations, and results visualized by graphics.

In my opinion, the paper should be structured according to the traditional approach (Introduction and Literature Review, Research Methodology, Results and Discussion, Conclusions).

The following comments should be solved.

  1. Analysis of fixtures for welding and their technological features should be discussed in the Introduction.
  2. What is the fixture? The fixture is the device that allows accurate locating and reliable clamping during the manufacturing process (for example, welding). Is it correct? Where are the fixture configurations in the presented manuscript? The fixture layout for analyzed variants should be presented and slightly described.
  3. The term “fixture” is used 10 times in the manuscript (1 in the title, 1 – in the abstract, 1 – in the keywords, 1 – in the main text, 6 – in the conclusions). In this regard, the title “Effects of the fixture configurations …” seems to be declarative.
  4. Please note that the titles for subchapters 2.1 and 2.3 are the same.
  5. Chapter 7 “Concluding remarks” should be changed to “Conclusions”.

Author Response

Dear Reviewer,

Thank you very much for your very useful suggestions and comments. We would like to stress that many of issues which you have mentioned we are doing now and will be describe in details in future. Our idea is considering weld metal made by two strength different weld metal where crack propagate from over-matching to under-match weld metal.

We have changed a text and description in paper according to your suggestions and marked here as DONE, as you can see.

Your comments:

  1. Analysis of fixtures for welding and their technological features should be discussed in the Introduction.  Parent plates during welding did not fixed
  2. What is the fixture? The fixture is the device that allows accurate locating and reliable clamping during the manufacturing process (for example, welding). Is it correct? Where are the fixture configurations in the presented manuscript? The fixture layout for analyzed variants should be presented and slightly described. DONE
  3. The term “fixture” is used 10 times in the manuscript (1 in the title, 1 – in the abstract, 1 – in the keywords, 1 – in the main text, 6 – in the conclusions). In this regard, the title “Effects of the fixture configurations …” seems to be declarative. 

We have put the text: Fixtures as device for accurate locating and reliable support during bending testing should be size according to ASTM E1820 standard. According to mentioned standard rollers should be free in order to keep constant loading arm. Fixed roller can have influence on results, as will discussed in this paper.

  1. Please note that the titles for subchapters 2.1 and 2.3 are the same.   DONE
  2. Chapter 7 “Concluding remarks” should be changed to “Conclusions”.  DONE

Kind regards

Nenad
